# Behavior of Ten Coffee Cultivars against *Hemileia vastatrix* in San Ramón (Chanchamayo, Peru)

**Ricardo Borjas-Ventura** [1],*[ID], **Leonel Alvarado-Huaman** [1][ID], **Viviana Castro-Cepero** [2][ID], **Diana Rebaza-Fernández** [3][ID], **Luz Gómez-Pando** [1][ID] **and Alberto Julca-Otiniano** [1][ID]

1 Departamento de Fitotecnia, Facultad de Agronomía, Universidad Nacional Agraria La Molina, Lima 12-056, Peru; lealvarado@lamolina.edu.pe (L.A.-H.); luzgomez@lamolina.edu.pe (L.G.-P.); ajo@lamolina.edu.pe (A.J.-O.)

2 Departamento de Biología, Facultad de Ciencias, Universidad Nacional Agraria La Molina, Lima 12-056, Peru; vcastro@lamolina.edu.pe

3 Departamento de Estadística e Informática, Facultad de Economía y Planificación, Universidad Nacional Agraria La Molina, Lima 12-056, Peru; dianarebaza@lamolina.edu.pe

* Correspondence: rborjas@lamolina.edu.pe

**Abstract:** The objective of this research was to know the behavior of ten coffee cultivars against *Hemileia vastatrix* in San Ramón (Chanchamayo, Peru). The test was carried out from April 2017 to March 2018 at the Coffee Germplasm Bank, at the Regional Development Institute "La Génova" (San Ramón, Chanchamayo) of La Molina National Agrarian University. 10 cultivars were selected (Catimor, Pache, Mundo Novo, Pacamara, Caturra, Catuaí, Bourbon, Typica, Maragogype and Geisha), each one with 5 plants (2 × 1 m) of seven years of age, installed under the shade of trees of *Inga* sp. (12 × 12 m). For the statistical analysis, it was worked as a Completely Random Design, considering each cultivar as a treatment and each plant as a repetition. The severity was evaluated throughout a year, in the lower, middle and upper third of each plant, using a graphic scale. For the months with high severity, a comparison of means obtained in each third of the plant and for each variety was made. Then, the severity means obtained in each cultivar were compared. The results showed that the severity was different in each of the ten cultivars evaluated. Throughout the year, the highest severity values corresponded to the months of April, May and June 2017. The severity was greater in the lower third and was decreasing until the upper third of the plant, presenting statistically significant differences. Maragogype, presented the highest severity (15.9%) and Pacamara the lowest (1.3%), values that were also statistically different.

**Keywords:** Peru; coffee rust; severity; Maragogype; Pacamara

## 1. Introduction

Coffee is one of the most important agricultural products in the world and has a great economic, social and environmental impact in producing countries. In Peru, there are more than 400,000 ha of coffee installed under the shade of trees and it has been cultivated for approximately 100 years. Historically, it has been the main agricultural export product and its cultivation is in the hands of approximately 200,000 families, but it is estimated that its production involves, directly and indirectly, two million Peruvians. One third of Peruvian coffee is exported as specialty coffees and stands out as the world's second largest producer of organic coffee [1].

The "coffee leaf rust" (*Hemileia vastatrix*) is considered the most important disease of coffee cultivation worldwide and the cause of important economic losses [2–4]. In Peru, it was reported for the first time in 1979 in the central jungle (Satipo) and for 40 years it has been present in coffee plantations [5].

In 2013, during the so-called "rust crisis", the damage reached levels of economic importance in all coffee-producing countries [6]. In Peru, the losses for the coffee sector were approximately 60% of the crop valued at about 290 million dollars. The epidemic was of such magnitude that it generated concern at all levels and led to the implementation of an emergency plan, with a fund of approximately 30 million dollars, managed by the National Agrarian Health Service (SENASA), the entity responsible for agrarian health in the country [7]. But the crisis revealed the lack of minimum technical information to develop an emergency plan and one of these shortcomings was knowledge about the behavior of commercial cultivars against this disease.

Worldwide, there are more than 200 commercial varieties, but very few varieties are cultivated in the world, especially in our continent [8]. In many coffee-growing countries, the strategy to improve coffee production and quality has been the introduction of varieties. Peru has not been the exception and various varieties have been introduced such as Typica, Caturra Roja and Caturra Amarilla, Pache, Catuaí, Maragogype, Catimor and others. But, this introduction is mostly done informally and then its behavior in the different coffee-growing areas is not evaluated, even less is known, its response to the main pests and diseases of the crop. In a survey of Peruvian coffee growers, 70% of those surveyed considered that Catimor is resistant to "coffee leaf rust", Catuaí 50%, Mundo Novo 30%, Typica 26%, Bourbon and Pache 25%, Caturra 22%. Maragogype, Pacamara and Geisha, it is cultivated by very few farmers; the first two are classified as susceptible and resistant. However, the third [9] shows an interesting behavior that needs to be confirmed. Therefore, this research work was carried out with the objective of determining the behavior of ten coffee cultivars against *Hemileia vastatrix* in San Ramón (Chanchamayo, Peru).

## 2. Materials and Methods

This experiment was carried out from April 2017 to March 2018 in the Coffee Germplasm Bank, installed in the Regional Institute for the Development of Forest "La Génova" of the Universidad Nacional Agraria La Molina (UNALM). It is located in the so-called central jungle of Peru, in the district of San Ramón, Chanchamayo province, Junín Region, at an altitude of 950 m and at 10°54′1″ S and 75°15′1″ W. The coffee plants were seven years old and spaced 2 × 1 m, under the shade of *Inga* sp. Trees. installed with a spacing of 12 × 12 m. The management of the plot mainly considered mechanical weed control, edaphic fertilization and no pesticides were applied to control pests or diseases.

The soil in the experimental area has a sandy loam texture, with an acid pH (5.7) and a calcium, magnesium and potassium content of 8.2, 1.23 and 0.34 $cmol_{(+)}$ $kg^{-1}$, respectively. The Cation Exchange Capacity (CEC) was 16.80 and the Percentage of Base Saturation (PSB) was 59%. The area has an average temperature of 24.8 °C, with a maximum of 30 °C and a minimum of 16 °C. Precipitation is approximately 1829 mm per year, irregularly distributed throughout the year, with excess precipitation in the months of December, January, February and March, and with water deficits in the months of June to August [10].

For this experiment, 10 coffee cultivars were selected (Table 1), each with 5 plants. The severity of the disease was evaluated over a year (March 2017 to April 2018), in the lower, middle and upper thirds of the plant, using the scale shown in Figure 1, established by Julca- Otiniano [7], who point out that incidence and severity are highly correlated. We must mention that *H. vastatrix* naturally appears with the beginning of the rains which occur during the end of the year (November and December).

For the statistical analysis of data, it was worked as if it were a Completely Random Design, considering each cultivar as a treatment and each plant as a repetition, that is, there was a test with 10 treatments and 5 repetitions. For the months with high severity (April, May and June) a comparison of means obtained in each third of the plant and for each variety was made. Then, the severity means obtained in each variety were compared. In all cases, the Scott-Knott test (95%) was used and the AGROSTAT [11] software was used.

**Table 1.** Coffee cultivars studied in this evaluation and collection site in Peru [9].

|   | Cultivar | Código BG | Origin |
|---|----------|-----------|--------|
| 1 | Catuaí | UNACAF-22 | Chanchamayo (Junín) |
| 2 | Geisha | UNACAF-26 | Chanchamayo (Junín) |
| 3 | Bourbon | UNACAF-50 | Chanchamayo (Junín) |
| 4 | Pache | UNACAF-115 | Lamas (San Martín) |
| 5 | Typica | UNACAF-119 | Moyobamba (San Martin) |
| 6 | Catimor | UNACAF-135 | Jaén (Cajamarca) |
| 7 | Pacamara | UNACAF-142 | Villa Rica (Pasco) |
| 8 | Caturra | UNACAF-150 | La Convención (Cusco) |
| 9 | Maragogype | UNACAF-170 | La Convención (Cusco) |
| 10 | Mundo Novo | UNACAF-175 | La Convención (Cusco) |

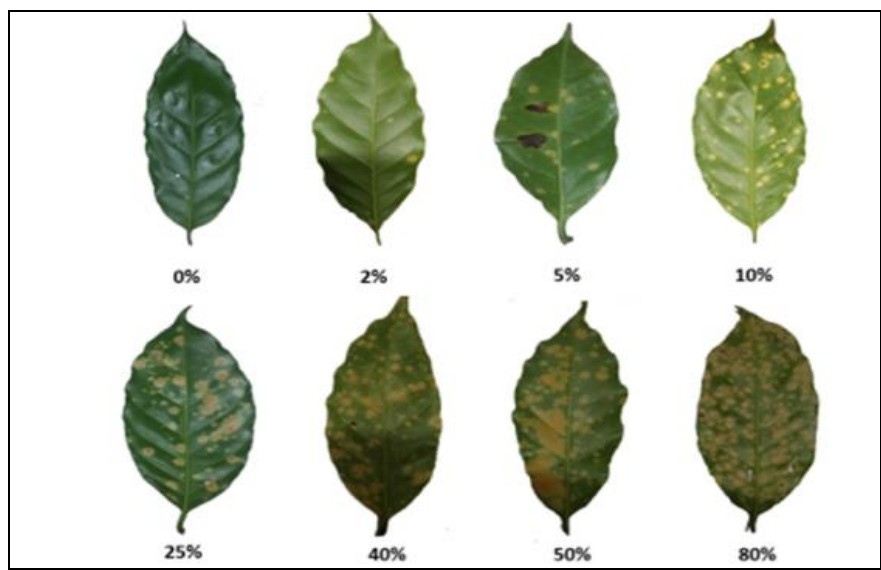

**Figure 1.** Degrees of severity of coffee leaf rust used in this experiment [7].

## 3. Results

The severity of "coffee leaf rust" in each of the ten evaluated cultivars changed over time, but the highest values were always recorded in the lower third of the plant (Figures 2–4). In cultivars such as Catimor and Pache, the disease in the lower third was recorded in the first three months of evaluation (April to June 2017), while in others such as Typica, Mundo Novo, Bourbon, Geisha and Catuaí, it was recorded in the entire study period, observing the lowest values between the months of August to October. In Pacamara, the severity was more irregular over time, but always with values lower than 10%. The highest values corresponded to Maragogype in the month of March 2017, above 30% severity (Figure 2).

In the middle third of the plant, the results showed a trend quite similar to that reported in the lower third. In cultivars such as Catimor and Pache, the disease was only registered in the first three months of evaluation, while in others such as Typica, Mundo Novo, Bourbon, Geisha and Catuaí, it was registered practically throughout the study period, observing the values lowest between the months of September to November (Mundo Novo), October to November 2017 (Bourbon), October to December 2017 (Typica), October 2017 to March 2018 (Maragogype). In Pacamara, it was only registered in the months of June and July 2017 and always with values lower than 10%. In this part of the plant, the highest value corresponded to Maragogype in April 2017, close to 20% severity (Figure 3).

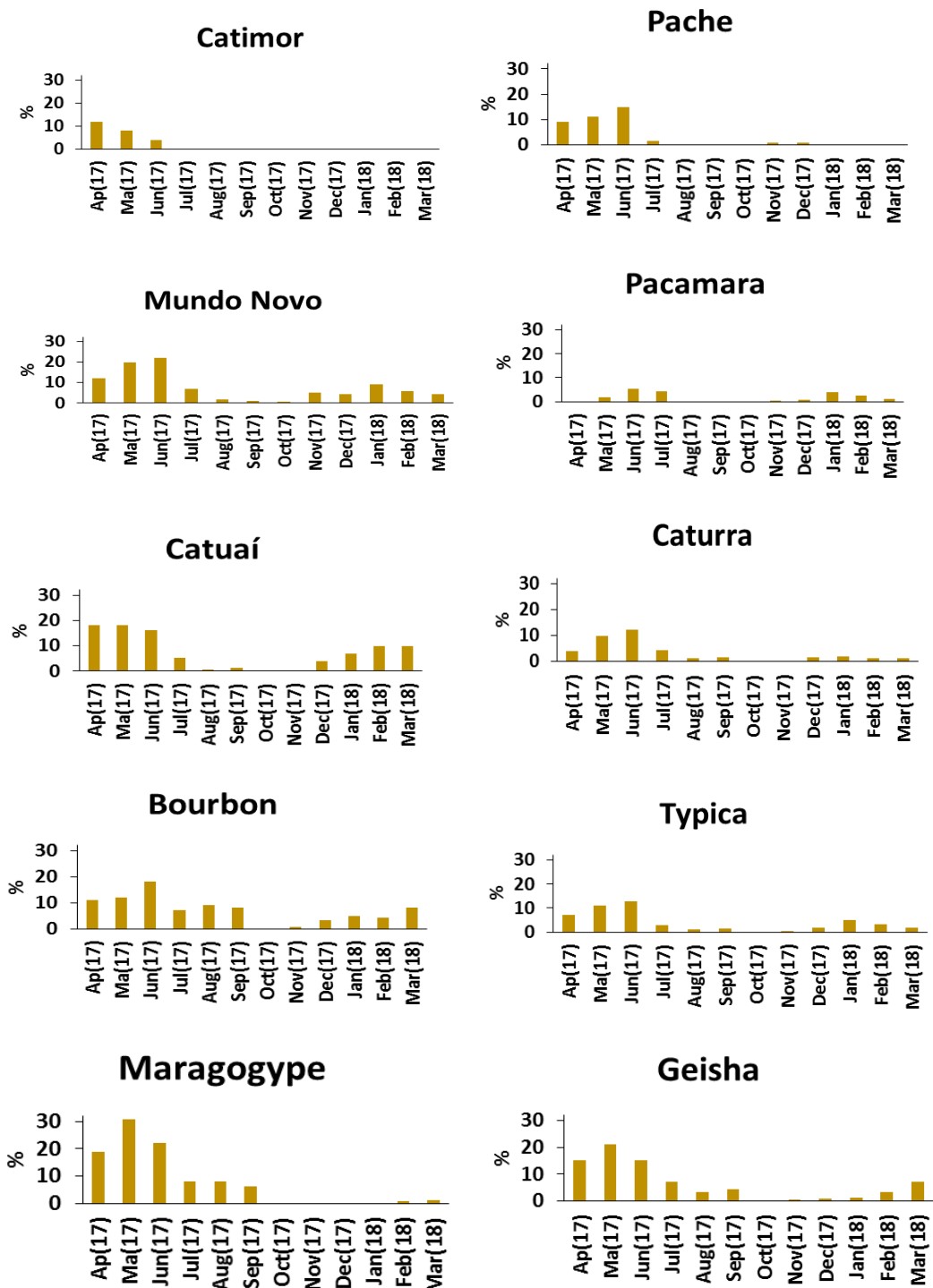

**Figure 2.** Coffee Leaf Rust dynamics, in each cultivar, from April 2017 to March 2018, in the lower section.

In the upper third of the plant, the presence of the disease was more irregular. For example, in cultivars such as Mundo Novo and Pacamara, the disease was registered in a single month of the year 2017; while in others such as Catimor and Typica, it was recorded in two months throughout the study period. In Maragogype and Bourbon, the severity was more irregular over time, but always with values lower than 10% (Figure 4).

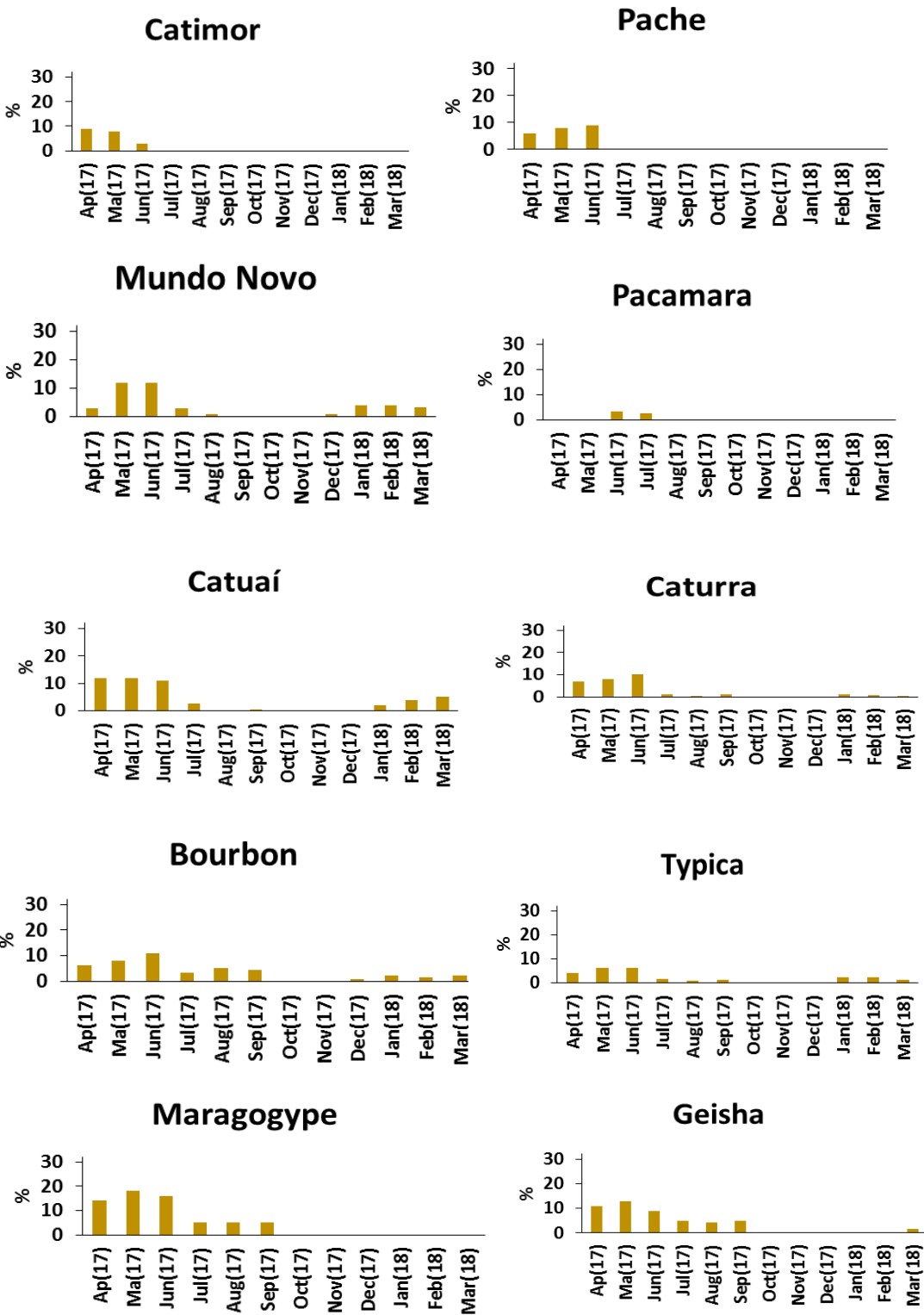

**Figure 3.** Coffee Leaf Rust dynamics, for each cultivar, from April 2017 to March 2018, in the middle section.

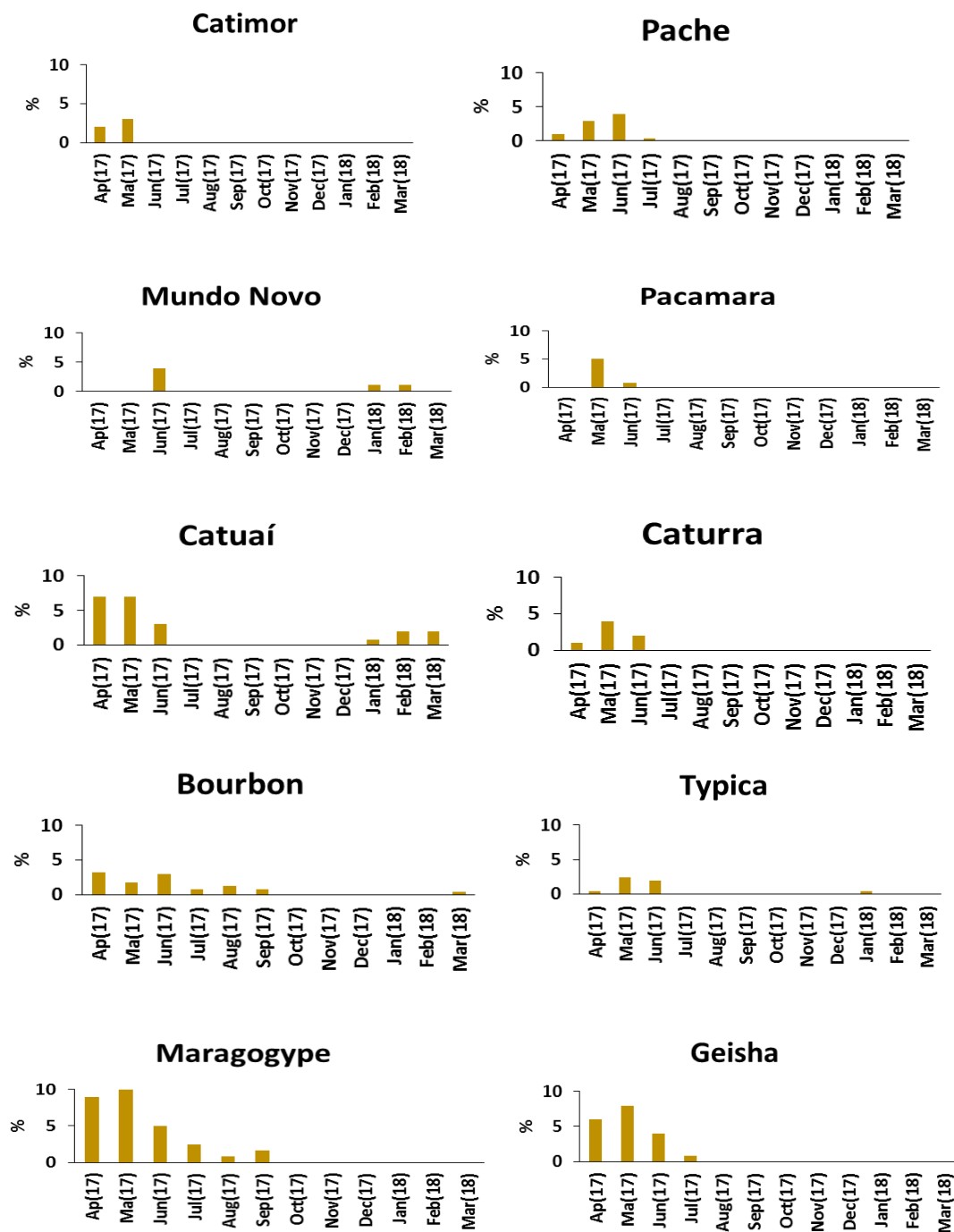

**Figure 4.** Coffee Leaf Rust dynamics, for each cultivar, April 2017 to March 2018, in the upper section.

When evaluating the average severity in each third of the plant, in all the cultivars studied, the differences in severity found in each level of the plant were statistically significant. Catimor did not have values that exceeded 10% of severity in the three levels studied; the same occurred in Caturra and Pacamara, while in Pache, the severity was greater than 10% in the lower third. In Mundo Novo, Catuaí and Geisha cultivars, severity in the lower third was close to 20% and in Maragogype a severity greater than 20% was found in the lower third (Figure 5).

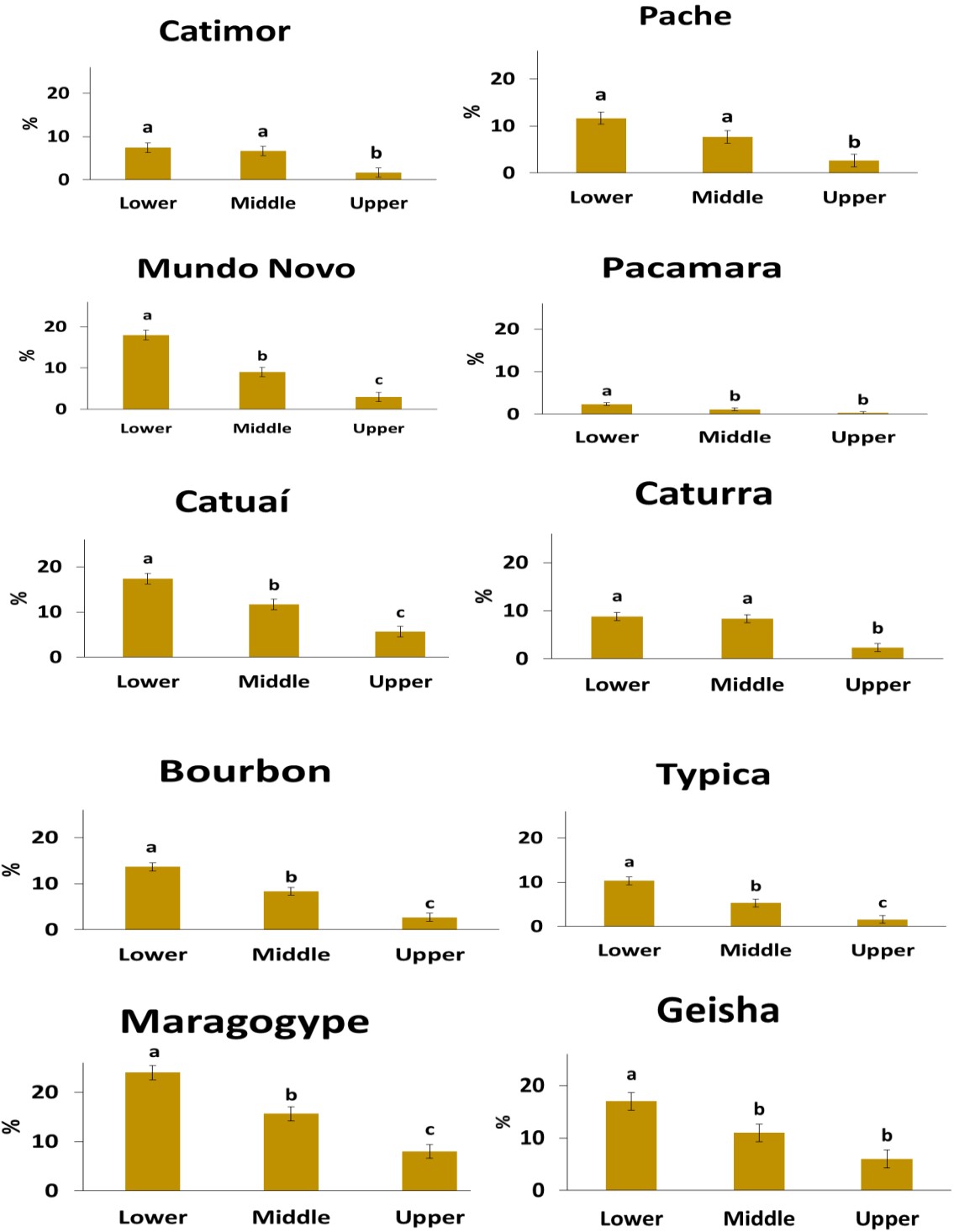

**Figure 5.** Coffee Leaf Rust (CLR) severity in the lower, middle, and upper sections for each cultivar. CLR severity values are the result of the April, May, and June average. Different letters indicate statistical differences according to the Scott-Knott test ($p \leq 0.05$).

The analysis of the average severity of each cultivar studied showed significant differences between them. The highest severity corresponded to Maragogype with 15.8%, a statistically higher value than those found in the other cultivars. The cultivar Mundo Novo, with a severity of 10%, was statistically similar to Catuaí and Geisha, which had values greater than 11%. Catimor, Pache, Caturra, Bourbon and Typica had values below 10% and were statistically similar, with Catimor

standing out with a value of approximately 5%. Pacamara was the cultivar in which the least severity of "coffee rust" was found, with a value of 1.3%, statistically different from the rest of the cultivars studied, as shown in Figure 6.

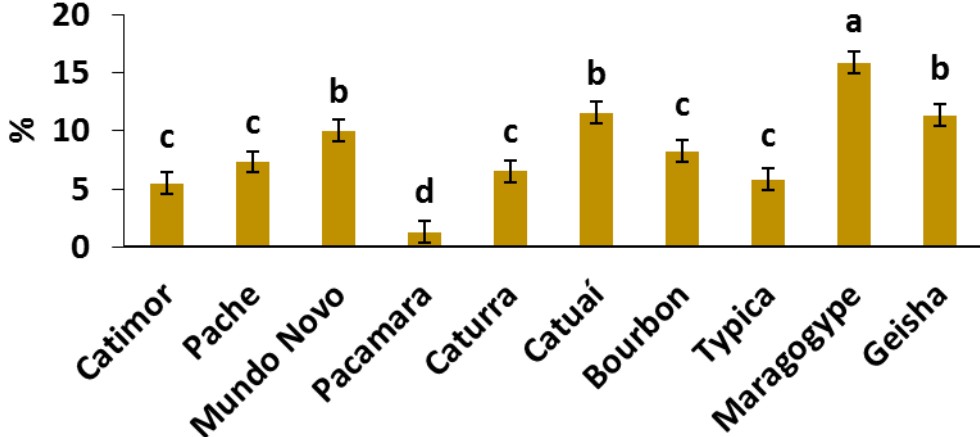

**Figure 6.** Coffee Leaf Rust severity in each cultivar in this experiment. The data shown are the result of averaging the lower, middle, and upper sections from April to June. Different letters indicate statistical differences according to the Scott-Knott test ($p \leq 0.05$).

## 4. Discussion

Most of the evaluated cultivars are of the traditional *Coffea arabica* or are the result of selection and natural mutations. According to SCAA [12] in the first group are Typica and Geisha, while Mundo Novo is a selection of Typica and Catuaí of Caturra; Maragogype and Pache, are mutations of Typica and Caturra of Bourbon. Pacamara is the result of the crossing of Pacas x Maragogype and Catimor of crossing Caturra x Híbrido de Timor. Julca-Otiniano [7] points out that, in Peru, the most widely distributed cultivars are Catimor, followed by Typica, Caturra, Pache and Bourbon. However, it clarifies that the wide distribution of a cultivar does not necessarily mean a greater area planted, since it is common to find more than one cultivar on the same farm, even mixed with each other in very small plots.

The variation in the severity of "coffee leaf rust" (CLR) over time in each of the ten cultivars evaluated is the result of the various factors that influence the development of the disease. And is that the severity of this disease not only depends on the cultivar used, but also depends on the geographic area, local climatic conditions, shade, humidity and the fruitful load of the coffee plant [5,13,14]. Zambolin [15] indicates that the infection is a function of altitude, temperature, precipitation, productive load, distance among plants, leaf humidity, relative humidity and the nutritional status of the plant.

In the study area, the time of greatest rainfall begins in October and ends in April of the following year. The greater severity of CLR in the months of April to June 2017 is mainly explained by coinciding with the harvest season in the area (950 m of altitude) which normally occurs between the months of April to June [16]. The incidence of this disease is higher at harvest time, probably because the phenolic compounds that protect the leaf migrate to the fruits [5]. This new distribution of compounds results in a lower defense of the plant [17]. It is also considered that, during harvest, the spread of the pathogen is greater (and therefore a higher incidence) due to the transit of harvesters from one plot to another or from one farm to another [5,18]. It has also been reported that "coffee rust" is correlated with fruit load [5,19] and this explains up to 50% of the variability of the disease [20]. López [18] observed a gradual increase in the disease as the number of productive nodes in the plant increased. Zambolin [15] points out that this disease follows a pattern of high and low intensity with high and low cherry load years, respectively. On the other hand, a high severity in a mostly dry period (May and

June) has been reported by Suresh [21] who found peaks with the highest incidence of CLR in the months with less precipitation.

A greater severity of CLR in coffee, in the lower third of the plant, in relation to the middle and upper third, has been reported in other research works. The lower part of the plants would have better conditions for the presence of the disease, that is, little light, low temperature and high humidity. Light negatively affects pathogen development [22]: high light intensity can affect its sporulation and latency period [23]. Furthermore, a suitable temperature (±24 °C), shade and humidity of the leaves can stimulate the infection process of *H. vastatrix* [2,22,23]. Zambolin [15] points out that the optimum temperature for the germination of the fungus spore is between 21 and 25 °C in the absence of direct light. According to Julca-Otiniano [24], the different incidence of diseases in each third of the plant would suggest that pathogens not only find favorable environmental and nutritional conditions in specific parts of the coffee tree, but also that there would be a competition between them to colonize. the foliage of the plant, as has been reported among *H. vastatrix*, *Mycena citricolor* and *Cercospora coffeicola* in Catimor, Colombia and Costa Rica 95, in Villa Rica, central Peruvian jungle.

The difference in severity of CLR between the cultivars studied is a response to their genetic differences, which have already been reported by other researchers [24–26]. Maragogype, Mundo Novo, Catuaí and Geisha, presented the highest severity values and were shown as susceptible to the disease, but with lower values than those reported in other investigations. Thus, in Catuaí, a severity of 84% has been reported [27]; while in Mundo Novo and Maragogype, 70 and 83% incidence, respectively [28]. According to the WCR [25], Typica, Bourbon, Caturra and Pache cultivars are susceptible to CLR; but in this experiment they had a medium severity (between 5.8 to 8.2%). This could be due, as previously stated, to the effect of other factors that influence the presence of CLR [13,14]. And it is that in other studies in these same cultivars higher levels of the disease have been reported. This is the case of Typica in Quillabamba, Cusco, where an incidence of 14.3% was achieved [29] and of Caturra Roja in Villa Rica where an incidence of up to 51.9% was reported [5]. Catimor had a severity of 5.5%, a value that is considered relatively high given that in previous years it had an average incidence of 0.08% [9]. In studies carried out in commercial fields on this cultivar, an incidence of 1.7% was reported in the Villa Rica area [24]. The cultivar Pacamara showed the lowest severity (1.3%), a result that suggests its resistance to *H. vastatrix*, although other investigations have indicated that this cultivar has a highly variable behavior [25], with low or moderate resistance to the disease [25,30].

Finally, another important factor to consider is the presence of different races of the pathogen, because the aggressiveness of the disease can vary due to the presence of one race or another [13] and many of the cultivars are improved to deal with a particular race. For example, Catimor was selected for its resistance to rust race II [18]. In Peru, there is no information on the rust races present in the different coffee-growing areas, but the hypothesis of the presence of several races is increasingly valid. Quispe-Apaza [31] indicate that races II and XXII are found in both Quillabamba and Villa Rica and that the Peruvian *H. vastatrix* populations present haplotypes similar to those of Colombia.

## 5. Conclusions

The severity of "coffee rust" was different in each of the ten cultivars evaluated. Throughout the year, the highest values corresponded to the months of April, May and June 2017. The severity was greater in the lower third and gradually decreased to the upper third of the plant; these differences were statistically significant. The cultivar Maragogype presented the highest severity (15.9%) and Pacamara the lowest (1.3%), values that were also statistically different.

**Author Contributions:** R.B.-V., L.G.-P. and A.J.-O. conceived and designed the experiments; L.A.-H. and V.C.-C. performed the experiments; D.R.-F. and R.B.-V. analyzed the data. All authors have read and agreed to the published version of the manuscript.

**Funding:** This research received no external funding.

**Conflicts of Interest:** The authors declare no conflict of interest.

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
