# Peer review of "Behavior of Ten Coffee Cultivars against Hemileia vastatrix in San Ramón (Chanchamayo, Peru)"

_agronomy, doi:10.3390/agronomy10121867_

Round 1

Reviewer 1 Report

The authors of this paper describe the different degree of severity of Coffee rust in ten cultivars of this plant. Even though they use advanced statistical techniques and a well-defined random design, the paper is a mere visual characterization about the course of a fungal disease in coffee plants.

I think that authors must review the following points in order to improve its quality:

Summary

Line 11: Please, write ‘Hemileia vastatrix’ in italics (Hemileia vastatrix).

Introduction

Line 61: Again, Hemileia vastatrix has to be written in italics.

Materials and Methods

In the whole section, I miss a clear explanation about how the fungus was introduced.

  • Was the fungus widely present in this soil?
  • How was the fungal presence confirmed?
  • Did de authors inoculated the plants with a fungal solution?
  • In that case, how was the solution prepared and inoculated?

Results

Be careful with figures’ title format. Only the header of figure 1 (line 97) is properly formatted. Besides, figure’s frames are unnecessary in my opinion. Please, remove them.

Line 312: In order to show the entire name of each cultivar, please make smaller the X axis ticks.

Discusion

Line 368: Letter H from H. vastatrix must be written in italics.

Line 395: ‘H.vastratix’ must be written in italics.

Reviewer 2 Report

In this Manuscript, authors have accessed the performance of ten coffee cultivars against the fungus Hemileia vastatrix. The study is informative but needs a better interpretation of the results. My suggestions are below

  1. I would like to see the dynamics of infection severity (%) and environmental conditions throughout the experimental time points. As the environmental factors play a crucial role in the disease progression, it would have been much informative if the monthly avg. temperature and precipitation data for each cultivar and corresponding severity were depicted in the same graph.
  2. The discussion section must also account for the role of environmental regimes, for example, the correlation between severity % with temperature/Precipitation of specific months.
  3. Please change the scale of figure 4. It does not make any sense to show the higher values (30%) if none of the individuals have that value nearby it.
  4. Name of the cultivar is Catua í or Catuaí, or just a typo in the table?
  5. In the title please italicize the name of the fungus.

Round 2

Reviewer 1 Report

Dear colleagues,

Although you have clearly pointed out that the fungus is present naturally in the soil, I strongly recommend some molecular testing to confirm its presence in further studies.
